# Introduction of Alternative Proteins for Health Professionals’ Diet: The Transtheoretical Model and Motivational Interviewing of Nutritional Interventions

**DOI:** 10.3390/ijerph20043097

**Published:** 2023-02-10

**Authors:** Iliana Tsampoula, Martha Kelesi, Ioanna V. Papathanasiou, Georgia Thanasa, Aspasia Goula, Dimitrios Papageorgiou, Evridiki Kaba

**Affiliations:** 1General University Hospital «AΤΤΙΚOΝ», 11635 Athens, Greece; 2Nursing Department, University of West Attica, 11521 Athens, Greece; 3Nursing Department, University of Thessaly, 41110 Larissa, Greece; 4General Hospital of Athens “G. Gennimatas”, 11527 Athens, Greece; 5Business Administration, University of West Attica, 11521 Athens, Greece

**Keywords:** alternative proteins, Transtheoretical Model, motivational interview, health professionals, nutrition

## Abstract

The World Health Organization has documented the impact that a balanced diet has on disease prevention. Overconsumption of meat can lead to obesity, arteriosclerosis, thrombosis, type 2 diabetes and many life-threatening diseases. A new group of proteins has currently emerged in the scientific community of alternative nutrition called “alternative proteins”. Several interventions have been introduced by a large number of healthcare providers in order to promote and enhance people’s dietary habits. Two of the most prevalent models of health-related behaviour modification are the transtheoretical—stages of change—model (TM) and motivational interviewing (MI). The purpose of this study is to research and examine how the ΤΜ and MI could be effectively implemented in the diet of health professionals through alterations of eating habits. The population of the study is to comprise health professionals from «AΤΤΙΚOΝ» University General Hospital, Athens, Greece. The sample of participants is to be selected by the professional environment of the researcher. Participants, through random selection, are to be divided into two groups: the control group, consisting of 50 individuals, and the intervention group, consisting of 50 individuals. The duration of the study is to be from November 2022 to November 2024. The present study is concerned with productive mixed—quantitative and qualitative—evolutionary research along with the application testing of both the ΤΜ and the MI. It is to be conducted specifically on health professionals via self-administered questionnaires and semi-structured interviews.

## 1. Introduction

The world is faced with a major issue of meat overconsumption and its effect on people’s health. In 2017, Americans were reported by the U.S. Department of Agriculture (USDA) to have exceeded the recommended meat consumption levels set by national dietary guidelines [1]. According to a risk factor analysis over the period 2010–2020, the leading causes of mental and physical disabilities, as well as early death, are associated with poor dietary habits in the United States [1]. The positive effect that a plant-based protein diet has had on the prevention of various diseases has been documented through research over the years [1,2]. Recent epidemiological studies from the United States and Europe indicate that overconsumption of red meat—particularly processed meat—is associated with an increased risk of total mortality rate, due to cardiovascular disease, colorectal cancer and type 2 diabetes in the long run for both sexes [2,3]. Owing to the increasing rate of diseases, public health strategies ought to be activated in order to provide quality nutritional dietary behaviours and reduce chronic disease risk.

A great number of interventions has been introduced by a lot of healthcare providers in order to promote and improve people’s dietary habits [4,5]. The ratification of any useful change of habit requires an alternative mindset and, consequently, a modifying behaviour. Plenty of reliable tools are currently available to detect and alter abnormal or dysfunctional behaviour [5,6]. The so-called “cognitive models” aim to examine the factors that contribute to new behavioural adjustment and, thus, a change in dietary approach [7]. Two of the most prevalent models of health-related behaviour modification are the transtheoretical—stages of change—model (TM) and motivational interviewing (MI) [5,6,7,8]. The scientific knowledge obtained is to highly contribute to the advancement and development of health professionals’ health related to the benefits of the alternative protein consumption. Furthermore, the TM and MI use is to be proven auspicious in diet modification.

### 1.1. Alternative Proteins

The global food market and the World Health Organization have been focused on overconsumption of meat and its impact on burdened climate due to human activity, animal abuse, and health issues that humanity is facing nowadays [2]. In the United States and around the world, cardiovascular disease is the leading cause of death. While the risk of developing cardiovascular disease, including heart attack and stroke, increases with age, other risk factors are influenced by lifestyle [1,2]. Certain kinds of behaviour which are known to improve cardiovascular health entail eating healthy food, such as fruit and vegetables, regular physical activity, and controlling high blood pressure, cholesterol and blood sugar levels [2,3]. Consumption of red meat, especially of mammalian origin, such as beef, pork, and lamb, as well as processed meat, which is transformed through salting, curing, fermentation, smoking, or other processes to enhance flavour or preservation, has rapidly been increasing worldwide. These trends can have major negative health and environmental consequences. Rising incomes and urbanization are driving a global dietary transition in which traditional diets are replaced by diets higher in refined sugars, refined fats, oils and meats. By 2050, these dietary trends, if unchecked, would be a major contributor to an estimated 80% increase in global agricultural greenhouse gas emissions from food production and to global land clearing. Global food production emits more than a quarter of all greenhouse gases (GHG), with livestock accounting for more than half of the emissions. Global food production also uses nearly 40% of the Earth’s land as cropland and pastureland. As for freshwater resources, it consumes more than two-thirds of the planetary freshwater from both surface water and groundwater, but it also pollutes water ecosystems with nitrogen and phosphorus. To achieve the goal of creating sustainable dietary systems that provide healthy diets for a growing population but simultaneously reduce environmental impacts, analyses of the quantitative linkages between diets, human and environmental health are of great significance [1,2,3]. Current research shows that there are certain chemicals in red and processed meats—both added and naturally occurring—that cause these foods to be carcinogenic. Considerable evidence from long-term prospective cohort studies has demonstrated that diets high in red and processed meat are associated with increased risk of type 2 diabetes (T2D), cardiovascular disease (CVD) and all-cause mortality [3,9]. The number of companies working on producing meat alternatives is growing worldwide. Recently, a new group of proteins which are called “alternative proteins” has emerged in the world of alternative nutrition. In addition to environmental awareness and economic benefits of consuming alternative proteins, their contribution to human health is very promising [3,9]. Plant-based proteins such as soy, lab-grown meat, single-cell proteins from yeast or algae, and edible insects are some of the alternative proteins [9] which are proven to be a great source of vitamins, fiber, minerals, as well as being low in cholesterol and fat [10,11,12]. Despite their proven nutritional value and the benefits that “alternative proteins” have on human health and environmental awareness, Western consumers reluctantly address them due to established nutritional profiles, lack of information or even increased costs [13,14]. In addition, people’s eating habits are shaped by cultural norms, specifically etiquette on consuming food, which is normally quite challenging to change [15]. The implementation of dietary solutions to the tightly linked diet–environment–health trilemma is a global challenge, and opportunity, of great environmental and public health importance.

### 1.2. The Transtheoretical Model through Stages of Change and the Motivational Interviewing as Interventions to Eating Behaviour Modifications

In the present study, the TM was chosen because of its diversity in relation to other theoretical approaches which aim at behavioural change due to the fact that the behavior modification is interpreted as a sequential processing in this particular model, which involves five stages (Figure 1): 1.2.1. The precontemplation stage: The person does not intend to change behaviour over the next six months. Those involved in this stage have never received information, are poorly informed about the consequences of their behaviour, or feel discouraged due to previous failed attempts.1.2.2. The contemplation stage: The person intends to change their behaviour over the next six months. They are conscious of the advantages and disadvantages of behaviour modification.1.2.3. The stage of preparation: The person intends to take action to modify their behaviour in the immediate future (more specifically, in the period of one month) and they have already formulated an action plan aimed at changing their eating behaviour.1.2.4. The stage of action: The person in this stage has already made modifications regarding their lifestyle within the last six months.1.2.5. The maintenance stage: The individual takes action to avoid the possibility of relapse while feeling progressively more confident about their ability to maintain the behavioural change already achieved [16,17,18,19].

The TM is a theoretical model of behavior change which has been the basis for the development of effective interventions aimed at promoting the change of health-related behaviors. It provides the theoretical framework for understanding the process of behavior modification and MI is the tool that facilitates this modification [16,17]. Progress and stage-change processes can be measured and even small steps of change can be reinforced and the outcome can be assessed [17]. The individual enters the thought process by evaluating the pros and cons associated with this modification. The TM is the theoretical model of behavior change that examines the way that a particular group of individuals behaves towards the problem they are faced with and investigates the possibility of modifying this behavior [18]. The application of the TM has been incorporated into the development of the MI [16]. The main points of MI according to Rollnick and Miller are as follows:Motivation to change is generated by the individual and not imposed by external factors. Motivational interviewing is based on identifying and mobilizing the individual’s internal values and elements in order to change behavior.It is not the therapist’s but the individual’s purpose to clearly and distinctly articulate and resolve their ambivalent feelings. The counselor’s purpose is to enhance the expression of both components of ambivalence and lead the individual to an acceptable resolution of the problem which supports change.Direct persuasion is not an effective method to resolve ambivalent feelings. Such methods have been shown to increase the individual’s resistance and reduce the likelihood of behavior modification.The style of intervention should be calm and eliciting through the counselor’s use of arguments, direct persuasion and aggressive countering.The counselor is direct in helping the individual to examine and resolve their ambivalent feelings.Whether the individual is ready for the desired behavior change will be determined by the interpersonal interaction between the counselor and the patient.The therapeutic relationship isakin to a partnership or companionship [5,7,8,16].

The steps to be followed during the motivational interviewing technique are as follows (Figure 2):Step 1—Engagement

The quality of the therapeutic relationship plays a decisive role on the outcome of treatment.

Step 2—Focus

In this stage, the therapist guides the client to explore a certain path of achievement in order to produce the desired changes.

Step 3—Incitement

At this stage, the therapist promotes the required actions to be followed and helps to overcome any obstacles such as doubt, constant debate and lack of trust.

Step 4—Planning

The last process of motivational interviewing is planning, which involves drawing a strategic plan on ways to achieve the desired change [5,6,7].

The parallel use of these two tools is also articulated by the creators of the MI technique, who argue that the use of the TM offers an understanding of the change process, while the MI offers the tools and all the constructive processes to achieve the modification of deviant behavior [19].

No interventions were found in the literature that modify nutritional behaviors by introducing alternative proteins by applying both the TM and the MI. Therefore, it seems that there is a research gap and an opportunity to study the implementation of these interventions in order to explore the effectiveness regarding nutritional behaviour.

### 1.3. Aim and Objectives of the Study

The purpose of this study is to explore the ways in which the ΤΜ and MI serve as interventions in changing nutritional behavior by introducing alternative proteins to health professionals’ diet.

The objectives of this study are:(i)to investigate the nutritional behavior of the study population regarding the introduction of alternative proteins;(ii)to improve their diet and compliance regarding the introduction of alternative proteins in their diet during their monitoring;(iii)to evaluate the effectiveness of the TM in supporting the modification of health behaviors and assess the intervention program, its design and quality characteristics via the study population after the end of the program;(iv)to compare the effectiveness of a short counseling/information intervention lasting 10 min and a behavior modification intervention with the application of the MI intervention in the form of MET (motivational enhancement therapy) of four sessions of semi-structured interviews and telephone intervention in the study population.

## 2. Methods and Analysis

### 2.1. Design

The present study is concerned with productive mixed—quantitative and qualitative—evolutionary research with application testing of the theory of the TM and MI in a specific case in a special population—health professionals—using self-administered questionnaires and semi-structured interviews.

### 2.2. Ethical Approval

The study protocol has been approved by the scientific board of the «AΤΤΙΚOΝ» University General Hospital, Athens, Greece, on the 7 June 2022 with protocol number 335-25/05/2022, and from University of West Attica, Institutional Review Board Statement, 20-01-23/3159/16-01-23.

### 2.3. Participants and Recruitment

The population of the study will include health professionals from the «AΤΤΙΚOΝ» University General Hospital, Athens, Greece. Healthcare professionals were chosen as the study population because of their daily involvement with patients. Thus, it is believed that if health professionals will be convinced of the nutritional value of alternative proteins, they will be able to educate their patients regarding the outmost health benefits of these products. The sample of participants will be at the professional researchers’ disposal, who will select them. The sample will meet the qualitative criteria that are to be set in order to be included in the study and there will be no additional restrictions regarding demographic, social or other characteristics. Participants will have voluntary participation in the program and free access to it and their data throughout the program. Upon entering the study, they will receive a code number and will be randomized to control or intervention groups via random selection. Participants will also be able to quit at any time they choose without disrupting the process. Participants will be randomly divided into two groups: the control group, consisting of 50 individuals, and the intervention group, consisting of 50 individuals as well. 

#### Exclusion Criteria

The participants who will be excluded from the study and the implementation of the program are those who:Have been diagnosed with mental illness or depression.Do not speak Greek.Have a health problem diagnosed, such as a cardiovascular condition or diabetes.Have an allergy or intolerance to any of the recommended products, such as alternative proteins.Are vegan or vegetarians.Are not healthcare professionals

### 2.4. Nutritional Intervention Based on the Transtheoretical Model and Motivational Interviewing

Two main categories of intervention will be applied to the study population:

#### 2.4.1. Control Group

In the first category of participants, who will be the control group, a program of modification of eating behaviour will be carried out by introducing alternative proteins. Those are to be conducted in the form of a single counselling interview—educational session—of informative character lasting 10 min, during which information is to be provided regarding the importance of diet and alternative proteins. The control group is to complete the classification questionnaire in the stages of change of the ΤΜ both at the beginning and at the end of the intervention. The purpose is to draw conclusions regarding the desire to change their eating habits by intervening according to the statement of desire provided by the tool. 

#### 2.4.2. Intervention Group

A complete program of modification of eating habits is to be carried out in the intervention group according to the principles of the ΤΜ and with the application of the MI, which will be in the form of ME. It will include counselling and support intervention in the form of four semi-structured interviews and telephone interviews lasting 40 min and 10 min, respectively, in total. The intervention group will be evaluated upon completion of the classification questionnaire in the stages of change of the TM, at the beginning and end of the intervention. 

The information concerning the intervention will be made known to the whole population and the distribution of the participants under study. The research is to be conducted between two categories using a randomized method. The participants will not have any information regarding which of the two groups they have joined as well as the differences between the groups—simple blind study—in order to avoid a systematic error during the study [20,21].

Due to ethics, the control team was selected to receive an informative session meeting in order to benefit from the intervention program.

All participants are to receive the same thorough information about the objectives, procedures and benefits of the intervention program in order to be motivated to participate in the same way, as well as ensure anonymity and confidentiality of all personal data and information resulting from their participation in the program, throughout the program and after it. All participants are to receive separate forms:Information and mobilization form concerning the intervention program, research that will be carried out, goals, procedures and expected benefits from their participation.Signed consent form before joining the program.Complaint form.

### 2.5. Study Tools

The tools that are to be used in the study concern:Knowledge and attitude questionnaire for health professionals regarding the alternative proteins.

Participants will be able to freely express their already existing knowledge and views on the matter of the alternative proteins through closed-ended type questions. The questionnaire will be completed by the entire population under study. The researcher will draw useful information about the profile of the participants.

2.Classification questionnaire in the stages of the Transtheoretical—Stages of Change—Model

The classification questionnaire in the stages of the transtheoretical—stages of change—model [5,6,7] is to be completed by the entire population who will be under study at the beginning and end of the intervention. 

The evaluation of the intervention group is to be carried out upon completion of the questionnaire at the beginning of each session and at the end of the intervention. The evaluation of the change concerning the behaviour of participants will be classified through the movement of the population from one stage of the model to another, for example, moving from the stage of contemplation to the stage of action. It is also to be used as an indicator of successful non-modification of eating behaviour by including functional foods and compliance with preventive controls.The questionnaire is also to be completed by the control group at the beginning and at the end of the intervention. This action aims to draw conclusions regarding the ability to change the pre-desired eating habits and aims to compare the behavioural response of the group.

In the control group, the completion of the questionnaire at the beginning and end of the intervention will be used to draw conclusions regarding the desire to modify the eating behaviour at the beginning and end of the intervention according to the desire to modify the behaviour provided by the tool, and to compare the results with the corresponding desire of the population of the intervention group to modify their behaviour. Have you tried to include functional foods in your daily life? (Yes or No) 

3.Semi-structured interviews

There will be two types of semi-structured interviews which correspond to the categories of the study population. A 30-min preparatory interview is to be conducted initially for both categories of the populations under study. 

A.A 10-min interview will have an educational and advisory character and will not follow any other specific methodological scheme. It is to concern the category of the population/control group.B.There are to be four 40-min interviews/meetings with 10-min intermediate telephone interviews according to the methodology of motivational enhancement therapy (MET) [16,17,18,19]. The interviews are to be held every 15 days with an intermediate telephone interview that will be determined in agreement with each participant and will aim to monitor and evaluate the effort to modify the behaviour of the participants and provide brief advice on issues that may arise.

The principles and techniques of motivational interviewing will be applied in order to support the mobilization of the individual to follow changes in their behaviour through the resolving of ambivalence, method of ejection and empowerment of self-efficacy, and the management of the individual’s resistance to change through empathetic dialectic [16,17,18,19].

4.Questionnaire for the evaluation of attitudes, acceptance and willingness of consumers to pay for innovative products [22,23].

The questionnaire will be provided at the beginning and at the end of the process to both groups of participants and will aim to record the change in the behaviour of participants—consumers towards innovative products such as alternative proteins. This questionnaire has been created and weighed in the context of a graduate study conducted by a student of the Agricultural University of Athens, Department of Rural Economics and Development, in order to record the attitude, acceptance and willingness of consumers to pay for innovative products. For the present study, written permission has been obtained by the author for its use [23]. 

## 3. Statistical Analyses

### 3.1. Data Entry in Statistical Program (SPSS)

Before entering the data into the statistical program, the registration forms will be counted and numbered so that it is possible to check any omission or incorrect entry at any point in this process. The data will then be entered into the statistical program, which will be used to perform the analysis. The data will be entered into the SPSS statistical program through the SPSS Data Entry program, which will define restrictions and rules for the correct entry of data and the avoidance of errors.

### 3.2. Data File Checking, Error Correction from Data Entry in SPSS (Editing)

In this step, the entered data will be checked and the errors that may have occurred during the data entry will be corrected so that there is the possibility of traceability at all stages of the research.

### 3.3. Initial Descriptive Data Analysis

Statistical analysis of the data will start from their simple descriptive analysis, from the tables that will emerge from the statistical program. Exactly which techniques will be used depends largely on the descriptive analysis of the data and the results obtained from this research stage. It is at the discretion of the researcher to change or add statistical techniques in order to reach the depth of analysis that the data allow.

### 3.4. Descriptive Data Analysis

The deeper statistical analysis of the data will be performed using inductive statistical methods. Qualitative variables will be expressed as percentages. Qualitative variables will be compared using the chi squared test or the Fischer test, as appropriate. In addition, the odds ratios with 95% confidence intervals will be calculated. For the requirements of the analysis, the statistical package SPSS version 22 will be used.

### 3.5. Quality Data Analysis

The collection and analysis of the resulting qualitative data will be assessed based on the method of content analysis [24]. Qualitative content analysis is a process designed to condense raw data into categories or topics based on valid conclusions and interpretations. This process uses inductive reasoning, from which issues and categories emerge from the data through the researcher’s careful examination and continuous comparison [23].

The data of the participants will be encrypted anonymously and confidentially and the main researcher and the person in charge of the study will have full control and access to them. All the necessary identification data of the participants and the sensitive personal data will be governed by the principle of minimization and limitation of the purpose and will be kept in a separate file from the research data in which it will be processed.

The separate file will be accessed by the lead researcher and the scientific officer and a password will be required. The complete files (research and files containing the personal data) will be kept on a computer accessible only to the lead researcher and the scientific officer of the study (bears electronic passwords) for a total period of five years and will then be destroyed according to the appropriate legislation.

## 4. Conclusions

The present study is expected to improve the nutritional behaviour of healthcare professionals by introducing alternative proteins with the application of both the TM and the MI methods. Specifically, the patients of the intervention group are expected to modify their eating behaviour through the introduction of alternative proteins. Moreover, the patients of the intervention group are expected to move to the next classification stage according to the TΜ as an indication of action and progress on behaviour modification. It is expected that there will be a statistically strong relationship between the ranking stage of behaviour change and the improvement of eating behaviour, as well as the description of a statistically significant advantage of the MI and, in particular, the short MET method regarding achieving effective support for dietary modification of the second category of proteins (intervention group) in relation to the provision of short-term counselling—educational support of one session in the control group.

## Figures and Tables

**Figure 1 ijerph-20-03097-f001:**
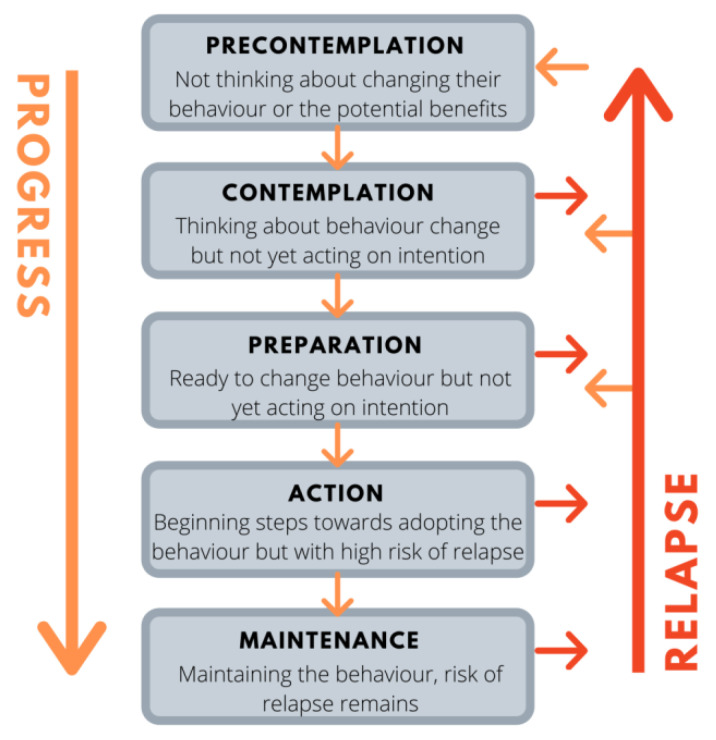
The transtheoretical—stages of change—model.

**Figure 2 ijerph-20-03097-f002:**
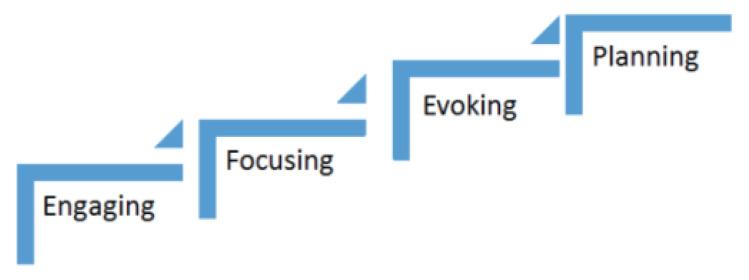
The steps to be followed during the motivational interviewing technique.

## Data Availability

Data sharing is not applicable to this article.

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
