# Peer review of "Introduction of Alternative Proteins for Health Professionals’ Diet: The Transtheoretical Model and Motivational Interviewing of Nutritional Interventions"

_ijerph, 2023, doi:10.3390/ijerph20043097_

Round 1
Reviewer 1 Report
General comments:
This protocol describes a planned mixed methods study to evaluate the efficacy of two methods of interventions on dietary consumption of alternative proteins. It is novel in applying these dietary intervention methods to the consumption of non-animal sources of protein, which is beneficial for both population health and the environment.
Below are suggestions to improve the protocol. The primary concern which requires major revisions is that the outcome variable – how the methods will be compared and evaluated – is unclear in the introduction and methods. However, I hope this issue can be fixed quite easily with one or two sentences added.
Major revisions:
Title
1. The title is quite difficult to understand on first reading. This could risk interested readers not finding the paper. I suggest the authors consider which are the essential components, and simplifying. For example “diet modification of health professional’s diet by introducing alternative proteins” could be rewritten as “increase in consumption of alternative proteins”.
Abstract
2. The abstract has many sentences on the background, and only the final sentence on the methods. This appears imbalanced for a protocol. The abstract will be more useful if it briefly includes: the study population, location, study duration, and primary outcome measured.
Introduction
3. It is not clear after reading the introduction what the primary result is. The objectives include comparing the effectiveness of various methods, but it doesn’t state how they will be compared. What is the key metric used to compare them? This should be included in the aims/objectives.
Methods
4. Section 3.4. Please give more details on the planned statistical analyses: which variables will be used in chi2 or fischer tests, which variables will be included in the odds ratio calculation, and what is the outcome variable of the odds ratio.
Minor revisions:
Introduction
5. “According to an analysis of risk factors over the period 2010-2020, the leading cause of early death and disability in the United States is diet.” Please include the reference, or state if it is the same as the last sentence.
6. Section 1.3. “No interventions, were found in the literature, that modify nutritional behaviors by introducing alternative proteins by applying both the TM and the MI. Therefore, it seems that there is a research gap and an opportunity to study the implementation of these interventions in order to study their effectiveness in modifying nutritional behavior”. This will be better if it comes after the definition of the two methods.
7. Section 1.4. There is repetition in sentence 2 which makes it difficult to read. The authors may also wish to consider numbering the objectives to make it easier to follow.
Methods
8. Could the authors please include how participants are randomized to control or intervention groups.
Author Response
We thank the reviewer for his/her thoughtful review of our work.
|
Reviewer No1 |
|
|
Major Revisions |
|
|
The title is quite difficult to understand on first reading. |
We thank the reviewer for her thoughtful review of our work. We have thoroughly re-reviewed the manuscript and corrected any errors we came across. The following is the corrected title:
«Introduction of alternative proteins for health-professionals’ diet: The transtheoretical model and motivational interviewing of nutritional interventions.»
|
|
The abstract has many sentences on the background, and only the final sentence on the methods. This appears imbalanced for a protocol. The abstract will be more useful if it briefly includes: the study population, location, study duration, and primary outcome measured. |
Thank you also the reviewer for pointing this out.
This is the new version:
The population of the study is to be comprised of health professionals, from «ΑΤΤΙΚΟΝ» University General Hospital, Athens, Greece. The sample of participants is to be selected by the professional environment of the researcher. Participants, through random selection, are to be divided into two groups: the control group consisted of 50 individuals and the intervention group of 50 individuals. Τhe duration of the study is to last from November 2022 to November 2024.
|
|
It is not clear after reading the introduction what the primary result is. The objectives include comparing the effectiveness of various methods, but it doesn’t state how they will be compared. What is the key metric used to compare them? This should be included in the aims/objectives. |
We thank the reviewer for this suggestion and we agreed. We added the above information :
The knowledge yielded will contribute to advancing and informing the health professionals about the importance and the benefits of the alternative proteins’ consumption also it will prove the beneficial use of the ΤΜ and the MI in diet modification.
The Precontemplation stage: The person does not intend to change behaviour over the next six months. Those involved in this stage have never received information, are poorly informed about the consequences of their behaviour, or feel discouraged due to previous failed attempts. The Contemplation stage: The person intends to change their behaviour over the next six months. They are conscious of the advantages and disadvantages of behaviour modification. The stage of Preparation: The person intends to take action to modify their behaviour in the immediate future, and more specifically in the period of one month, while they have already formulated an action plan aimed at changing their eating behaviour. The stage of Action: The person in this stage has already made modifications regarding their lifestyle within the period of the last six months. The Maintenance stage: The individual takes action to avoid the possibility of relapse while feeling progressively more confident about their ability to maintain their behavioural change already achieved. [16-19] Progress and stage change processes can be measured and even small steps of change can be reinforced and the outcome can be assessed.[17 ]. The main points of MI according to Rollnick & Miller, are as follows: • Motivation to change is generated by the individual and not imposed by external factors. Motivational Interviewing is based on identifying and mobilizing the individual's internal values and elements in order to change behavior. • It is not the therapist’s, but the individual's purpose to clearly and distinctly articulate and resolve their ambivalent feelings. The counselor's purpose is to enhance the expression of both components of ambivalence and lead the individual to an acceptable resolution of the problem, which supports change. • Direct persuasion is not an effective method to resolve ambivalent feelings. Such methods have been shown to increase the individual's resistance and reduce the likelihood of behavior modification. • The style of intervention should be calm and eliciting through counselor’s use of arguments, direct persuasion and aggressive countering. • The counselor is direct in helping the individual to examine and resolve their ambivalent feelings. • Whether the individual is ready for the desired behavior change will be determined by the interpersonal interaction between the counselor and the patient. • The therapeutic relationship is like a partnership or companionship.[12,14-16]
The steps to be followed during the Motivational Interviewing technique are as follows: Step 1 - Engagement The quality of the therapeutic relationship plays a decisive role on the outcome of treatment. Step 2 - Focus In this stage the therapist guides the client to explore a certain path of achievement in order to produce the desired changes. Step 3 - Incitement At this stage the therapist promotes the required actions to be followed and helps to overcome any obstacles such as doubt, constant debate and lack of trust. Step 4 - Planning The last process of motivational interviewing is planning, which involves drawing a strategic plan on ways to achieve the desired change. [12-14] |
|
Section 3.4. Please give more details on the planned statistical analyses: which variables will be used in chi2 or fischer tests, which variables will be included in the odds ratio calculation, and what is the outcome variable of the odds ratio. |
We thank the reviewer for pointing it out.
According to the statistician this will be decided when the first questionnaires are collected. |
|
Minor Revisions |
|
|
“According to an analysis of risk factors over the period 2010-2020, the leading cause of early death and disability in the United States is diet.” Please include the reference, or state if it is the same as the last sentence. |
We have modified the mistake pointed out by the reviewer. According to an analysis of risk factors over the period 2010-2020, the leading cause of early death and disability in the United States is diet [1]. |
|
Section 1.3. “No interventions, were found in the literature, that modify nutritional behaviors by introducing alternative proteins by applying both the TM and the MI. Therefore, it seems that there is a research gap and an opportunity to study the implementation of these interventions in order to study their effectiveness in modifying nutritional behavior”. This will be better if it comes after the definition of the two methods. |
We have modified the mistake pointed out by the reviewer and we added this information after the definition of the two methods. |
|
Section 1.4. There is repetition in sentence 2 which makes it difficult to read. The authors may also wish to consider numbering the objectives to make it easier to follow. |
We have modified the mistake pointed out by the reviewer.
The objectives of this study are: i) to investigate the nutritional behavior of the study population regarding the introduction of alternative proteins ii) to improve their diet and compliance regarding the introduction of alternative proteins in their diet during their monitoring. iii) to evaluate the effectiveness of the TM in supporting the modification of health behaviors and assess the intervention program, its design and quality characteristics via the study population after the end of the program. iv) to compare the effectiveness of a short counseling / information intervention lasting 10 minutes and a behavior modification intervention with the application of the MI Intervention in the form of MET (Motivational Enhancement Therapy) of four sessions of semi-structured interview and telephone intervention in the study population. |
|
Could the authors please include how participants are randomized to control or intervention groups. |
We have modified the mistake pointed out by the reviewer. We added the following paragraph:
Participants will have voluntary participation in the program and free access to it and their data throughout the program. Upon entering the study, they will receive a code number and will be randomized to control or intervene groups via random selection. |
|
The whole text received a deep linguistic readjustment |
|
Reviewer 2 Report
This manuscript evaluated if nutritional counseling (in particular TM and MI) can serve as interventions in changing nutritional behavior by introducing alternative proteins to the diet of health professionals.
Although this protocol could be of interest, there are important methodological errors so, in my opinion, it cannot be accepted in the presnet form. In particular, there are the following major problems:
- questionnaire construction
- Diet and nutritional counseling (TM and MI are counseling methods) should be managed by a dietician or a nutritionist. I cannot undestand why this figure was not involved in the study.
- Substantial english and punctuation revision is required.
Here there are some specific comments:
Abstract:
Lines 27-29: this phrase is too long and not clear. Please modify it.
Introduction:
line 35: The world is faced a major problem - is facing or has faced? Please modify
Line 36: In 2017, the U.S. Department of Agriculture (USDA) reported that Americans are exceeding the amount of meat recommended. It means that since 2017 Americans are exceeding? Please correct the verb tense or the adverb.
Line 54: I would not define "enormous" the contribution of alternative proteins to human health on the basis of only two studies. Certainly it is promising.
Lines 57-59: too difficult to read. Please modify the punctuation.
In my opinion the "The role of nutritional interventions" paragraph is useless, it can be added to the following one, as introduction.
Lines 77-79: these lines are difficult to read and understand. TM was chosen becaused, on the contrary to the other approaches that ameid at behavioural change, TM interpretes behavior modification as a sequential process?
Study design: study protocol. What you wrote was not a specification of the study design, moreover the phrase is very confunding. Please explain better
Knowledge and attitude questionnaire for health professionals concerning the alternative proteins. - Will you use open questions for the questionnaire? It is animportant methodological error, because of the possible great variability of the answers and, so, the difficulty in analysing results
Classification questionnaire in the stages of The Τranstheoretical - Stages of Change Μode - Is it a validated questionnaire? If yes, please insert the reference. If no, please clarify how will you develop it. MOreover, it is not clear how you will use it in the intervention group, please reformulate the phrase.
Author Response
We thank the reviewer for his/her thoughtful review of our work.
|
Reviewer No2 |
|
|
Major Revisions |
|
|
Diet and nutritional counselling (TM and MI are counselling methods) should be managed by a dietician or a nutritionist. I cannot understand why this figure was not involved in the study. |
In the nursing departments in all universities here in Greece there is a course Nutrition - Dietetics. Dieticians who are working at ATTIKON hospital will also take part in the study as health professionals and will be able to give advice if needed. As a nurse researcher I am trained to work the TM and MI as counselling methods for patients. |
|
Abstract:
Lines 27-29: this phrase is too long and not clear. Please modify it. |
We have modified the mistake pointed out by the reviewer. We deleted the unclear sentence. |
|
Introduction:
line 35: The world is faced a major problem - is facing or has faced? Please modify
Line 36: In 2017, the U.S. Department of Agriculture (USDA) reported that Americans are exceeding the amount of meat recommended. It means that since 2017 Americans are exceeding? Please correct the verb tense or the adverb.
Line 54: I would not define "enormous" the contribution of alternative proteins to human health on the basis of only two studies. Certainly it is promising.
Lines 57-59: too difficult to read. Please modify the punctuation. |
We have modified the mistake pointed out by the reviewer.
The world is faced with a major issue of meat overconsumption and its effect on people's health.
In 2017, Americans were reported by the U.S. Department of Agriculture (USDA) to have exceeded the recommended meat consumption levels set by national dietary guidelines [1] .
In addition to environmental awareness and the economic benefits of consuming alternative proteins, their contribution to human health is very promising [3-4].
Furthermore, livestock contributes to greenhouse gas emissions [6], because it represents 14.5% of all anthropogenic Green House Gas (GHG) emissions. Therefore, alternative proteins contribute to the desired climate change protection [5-7]. |
|
In my opinion the "The role of nutritional interventions" paragraph is useless, it can be added to the following one, as introduction. |
We transfer it in the introduction section. |
|
Lines 77-79: these lines are difficult to read and understand. TM was chosen because, on the contrary to the other approaches that aimed at behavioural change, TM interprets behaviour modification as a sequential process? |
We changed it according to the review comment as follows: The Precontemplation stage: The person does not intend to change behaviour over the next six months. Those involved in this stage have never received information, are poorly informed about the consequences of their behaviour, or feel discouraged due to previous failed attempts. The Contemplation stage: The person intends to change their behaviour over the next six months. They are conscious of the advantages and disadvantages of behaviour modification. The stage of Preparation: The person intends to take action to modify their behaviour in the immediate future, and more specifically in the period of one month, while they have already formulated an action plan aimed at changing their eating behaviour. The stage of Action: The person in this stage has already made modifications regarding their lifestyle within the period of the last six months. The Maintenance stage: The individual takes action to avoid the possibility of relapse while feeling progressively more confident about their ability to maintain their behavioural change already achieved. [16-19] Progress and stage change processes can be measured and even small steps of change can be reinforced and the outcome can be assessed.[17] The main points of MI according to Rollnick & Miller, are as follows: • Motivation to change is generated by the individual and not imposed by external factors. Motivational Interviewing is based on identifying and mobilizing the individual's internal values and elements in order to change behavior. • It is not the therapist’s, but the individual's purpose to clearly and distinctly articulate and resolve their ambivalent feelings. The counselor's purpose is to enhance the expression of both components of ambivalence and lead the individual to an acceptable resolution of the problem, which supports change. • Direct persuasion is not an effective method to resolve ambivalent feelings. Such methods have been shown to increase the individual's resistance and reduce the likelihood of behavior modification. • The style of intervention should be calm and eliciting through counselor’s use of arguments, direct persuasion and aggressive countering. • The counselor is direct in helping the individual to examine and resolve their ambivalent feelings. • Whether the individual is ready for the desired behavior change will be determined by the interpersonal interaction between the counselor and the patient. • The therapeutic relationship is like a partnership or companionship.[12,14-16]
The steps to be followed during the Motivational Interviewing technique are as follows: Step 1 - Engagement The quality of the therapeutic relationship plays a decisive role on the outcome of treatment. Step 2 - Focus In this stage the therapist guides the client to explore a certain path of achievement in order to produce the desired changes. Step 3 - Incitement At this stage the therapist promotes the required actions to be followed and helps to overcome any obstacles such as doubt, constant debate and lack of trust. Step 4 - Planning The last process of motivational interviewing is planning, which involves drawing a strategic plan on ways to achieve the desired change. [12-14] |
|
Study design: study protocol. What you wrote was not a specification of the study design, moreover the phrase is very confunding. Please explain better
|
We changed it as it follows :
The present study is a mixed method evolutionary research with the application of the TM and MI in a special population (health professionals) using self-administered questionnaires and semi-structured interviews. |
|
Knowledge and attitude questionnaire for health professionals concerning the alternative proteins. - Will you use open questions for the questionnaire? It is an important methodological error, because of the possible great variability of the answers and, so, the difficulty in analysing results |
We will use closed type questions.
Participants will be able to freely through closed-ended type questions express their al-ready existing knowledge and views over the matter of the alternative proteins. |
|
Classification questionnaire in the stages of The Τranstheoretical - Stages of Change Model - Is it a validated questionnaire? If yes, please insert the reference. If no, please clarify how will you develop it.
Moreover, it is not clear how you will use it in the intervention group, please reformulate the phrase. |
We insert the reference.
The classification questionnaire in the stages of The Τranstheoretical - Stages of Change Μodel, [12-14], will be completed by the entire population who will be under study at the beginning and end of the intervention.
We have modified the mistake pointed out by the reviewer. |
|
The whole text received a deep linguistic readjustment |
|
Reviewer 3 Report
Dear authors, thank you for the opportunity to review your Study protocol-
Please provide the sample size calculation. Are you sure that the sample of 50 subjects per group is sufficient for your conclusions to be valid and reliable?
Information about the assessment of the validity and reliability of the questionnaires and their reference is required
Author Response
We thank the reviewer for his/her thoughtful review of our work.
|
Reviewer No3 |
|
|
Please provide the sample size calculation. Are you sure that the sample of 50 subjects per group is sufficient for your conclusions to be valid and reliable?
|
Τhe statistician who has undertaken the support of the study informed us that the sample is sufficient as it concerns a prototype study that has both qualitative and quantitative characteristics. |
|
Information about the assessment of the validity and reliability of the questionnaires and their reference is required |
We inserted the reference.
In the reference 12-16 all the information about the validity and reliability are mentioned as the questionnaires are already tested and used in previous publications. |
|
The whole text received a deep linguistic readjustment |
|
Reviewer 4 Report
This study protocol aims to explore whether and how the TM and MI can work as interventions in changing eating behavior by introducing alternative proteins. This would be an interesting and of significance study as the "planetary health diet" has already been a hot topic.
This study protocol needs a revision before being accepted. The detailed comments are as follows:
1. Exclusion criteria in the section 2.3 (line 137): The authors are suggested to think about whether the vegetarians and the followers with some religions need to be excluded. The followers of some religions eat less/no meats.
2. TM and MI need to be introduced and illustrated with more details in the section 1.2 or 1.3.
3. In the section 1.1, the authors are suggested to discussion more about how over-consumption of red meats poses great pressures on both human health and environmental health. This would highlight the significance of your proposed study. The following literature provides such strong evidence and therefore is recommended to the authors -- "Trade-off between human health and environmental health in global diets" published in Resources, Conservation & Recycling. According to this article, the countries with over-consumption of red meats, such as US, are suffering high risks of diabetes and colorectal cancer, as well as high pressures of land use, water deletion, GHG emissions, etc.
Author Response
We thank the reviewer for his/her thoughtful review of our work.

Round 2
Reviewer 2 Report
For a more clear presentation of the steps of MI (and eventually TM), if possibile, I suggest the authors to use tables or figures.
Englire still require moderate revision
That's for my personal curiosity, you said that "Health care professionals were chosen as the specimen of the study because of their daily involvement with patients". If you will have a positive result of this study and health professionals will encourage patients to consume alternative proteins, do they will still use TM and MI? If yes, will they be instructed to do this?
Author Response
We want to thank the reviewer gor his/her thoughtful review of our work.

Reviewer 4 Report
The authors have addressed some of my comments. For the exclusion criteria, I agree with the authors that the religion issue cannot be asked because it is a sensitive question.
The following issues still need to be addressed:
1. In lines 70-71, the authors claimed that "these trends have major negative health and environmental consequences [1-3]". The cited references (1-3) cannot fully support this statement as these are health-related studies but provide no strong evidence on environmental impacts. The authors need to reinforce their statement by providing more references that focus on both human and environmental issues in global diets such as follows:
Trade-off between human health and environmental health in global diets. Resources, Conservation & Recycling 182, 106336 (2022)
Global diets link environmental sustainability and human health. Nature 515, 518-522 (2014)
2. In line 337, citation[25] is missing in the reference list.
Author Response
We want to thank the reviewer for his/her thoughtful review of our work
